# Squeeze Flow of Stress Power Law Fluids

**Lorenzo Fusi** *,† and **Andrea Ballotti** †

Dipartimento di Matematica e Informatica "U. Dini", Viale Morgagni 67/a, 50134 Firenze, Italy;
andrea.ballotti@stud.unifi.it
* Correspondence: lorenzo.fusi@unifi.it
† These authors contributed equally to this work.

**Abstract:** In this paper, we studied the squeeze flow between circular disks of a new class of fluids defined by an implicit relation referred to as stress power law fluids. The constitutive response of these fluids was written expressing the symmetric part of the velocity gradient as a tensorial function of the Cauchy stress. We assumed that the aspect ratio between the gap separating the disks and the radius was small so that a lubrication expansion could be adopted. We wrote the general problem and looked for a solution that could be written in terms of the small aspect ratio parameter. We obtained a sequence of problems that could be solved iteratively at each order, and we focused on the leading and first order, deriving explicit expressions for the velocity field, stress, and pressure.

**Keywords:** squeeze flow; power stress fluids; asymptotic expansion; lubrication flow




## 1. Introduction

Recently, Malek et al. [1] studied incompressible fluids from a new perspective, namely that in which the constitutive equation is written expressing kinematical quantities as a function of the stress. This is in contrast with the general approach adopted by Noll [2] to describe the response of incompressible simple fluids, wherein the Cauchy stress is determined by the history of the deformation gradient. The new approach provides the appropriate way to express fluids such as the yield stress fluids or fluids with rheological parameters that depend on the pressure. The new class of fluids introduced in [1], that is termed *stress power law fluids*, includes the classical incompressible Navier–Stokes model as a special subclass. An important feature of stress power law models is that they automatically satisfy the incompressibility constraint without requiring the definition a Lagrange multiplier to enforce the constraint. In the stress power law model, the constitutive equation is defined by the symmetric part of the velocity gradient being a function of the deviatoric part of the Cauchy stress tensor.

This new class of fluids has many interesting applications, among which we must mention blood and colloidal dispersions; see [3–6]. Blood is indeed a very complex mixture of a liquid plasma, red blood cells, platelets, proteins, ions, and other components that in large vessels can be safely modeled as a single-component fluid. Many biochemical reactions that occur during blood flow result in significant mechanical effects that may lead, for instance, to a reduction of the shear rate for increasing shear stress (coagulation) or to an increase of the shear rate for increasing shear stress (disaggregation of red blood cells). This type of behavior may occur in different regions of the shear stress/shear rate curve, exhibiting thus a non-monotonic relation between the shear rate and the shear stress that can only be described by implicit constitutive laws.

The constitutive equation of a generalized Stokesian fluid is such that the Cauchy stress tensor is given as a tensorial function of the density $\rho^*$ and of the symmetric part of the velocity gradient (starred quantities denote dimensional variables). $\mathbb{D}^* = 2^{-1}(\nabla v^* + \nabla v^{*^T})$, namely:

$$\mathbb{T}^* = \mathbb{F}^*(\rho^*, \mathbb{D}^*). \tag{1}$$

For a generic Navier–Stokes fluid, such an equation acquires the form:

$$\mathbb{T}^* = -p^*(\rho^*)\mathbb{I} + \lambda^*(\rho^*)\Big(\mathrm{tr}\mathbb{D}^*\Big)\mathbb{I} + 2\mu^*(\rho^*)\mathbb{D}^*, \tag{2}$$

where $\lambda^*$ and $\mu^*$ are the so-called bulk viscosity and shear viscosity and where $p^*$ is the pressure. Relation (2) expresses the Cauchy stress tensor $\mathbb{T}^*$ as a function of the symmetric part of the velocity gradient $\mathbb{D}^*$. Taking the trace operator on both sides of (2), we find:

$$\mathrm{tr}\mathbb{T}^* = -3p^*(\rho^*) + \Big(2\mu^*(\rho^*) + 3\lambda^*(\rho^*)\Big)\mathrm{tr}\mathbb{D}^*, \tag{3}$$

which expresses the linear relation between $\mathrm{tr}\mathbb{T}^*$ and $\mathrm{tr}\mathbb{D}^*$, provided $2\mu^* + 3\lambda^* \neq 0$. Exploiting (3), the constitutive relation (2) can be rewritten as:

$$\mathbb{D}^* = \frac{1}{2\mu^*(\rho^*)}\left[\mathbb{T}^* - \frac{\lambda^*(\rho^*)\mathrm{tr}\mathbb{T}^* - 2\mu^*(\rho^*)p^*(\rho^*)}{2\mu^*(\rho^*) + 3\lambda^*(\rho^*)}\mathbb{I}\right], \tag{4}$$

where $\mathbb{D}^*$ is a function of $\mathbb{T}^*$, so that:

$$\mathbb{D}^* = \mathbb{H}^*(\rho^*, \mathbb{T}^*). \tag{5}$$

In general, a constitutive equation of type (1) cannot be inverted to one of type (5) and the opposite is also true. In an even more general framework, one can assume that the constitutive relation is fully implicit, i.e.,

$$\mathbb{G}^*(\mathbb{T}^*, \mathbb{D}^*, \rho^*) = 0 \tag{6}$$

In the paper [1] Malek et al. introduced a new class of fluid models wherein the tensor $\mathbb{D}^*$ has a power law relationship with the tensor $\mathbb{T}^*$. They named this new class of fluids *stress power law fluids*. The constitutive equation for such fluids is of the type (5) and cannot be inverted in an explicit manner to obtain an expression such as (1). In particular, they proved that for special values of the constitutive parameters, the relation between the norm of the tensor $\mathbb{D}^*$ and the norm of the tensor $\mathbb{T}^*$ can be non-monotone. Simple flows of stress power law fluids were studied in [7–12].

Let us consider an incompressible stress power law fluid wherein the Cauchy tensor (here, $p^*$ is the Lagrange multiplier due to the incompressibility constraint (pressure); see [13] for a discussion on the notion of pressure) $\mathbb{T}^* = -p^*\mathbb{I} + \mathbb{S}^*$ is such that $\mathbb{S}^*$ is related to $\mathbb{D}^*$ via:

$$\mathbb{D}^* = \alpha^*\left[\left(1 + \frac{\beta^*}{2}\,\mathrm{tr}\,\mathbb{S}^{*2}\right)^n + \gamma\right]\mathbb{S}^*, \tag{7}$$

where $\alpha^* > 0$ has the dimension of the inverse of a viscosity and $\beta^*$ has the dimension of the inverse of a squared pressure. The flow index $n$ is a real number, while $\gamma \geqslant 0$ is non-dimensional. Notice that for $(n, \gamma) = (0,0)$ or for $(\beta^*, \gamma) = (0,0)$, the classical Newtonian model with viscosity $(2\alpha^*)^{-1}$ is recovered. Taking the Frobenius norm on both sides of (7), we obtained:

$$|\mathbb{D}^*| = \alpha^*\left[\left(1 + \frac{\beta^*}{2}|\mathbb{S}^*|^2\right)^n + \gamma\right]|\mathbb{S}^*|, \tag{8}$$

where $|\mathbb{D}^*|^2 = \mathbb{D}^* \cdot \mathbb{D}^* = \mathrm{tr}\mathbb{D}^{*2}$. Relation (8) can be plotted in the first half-quarter of the $\mathbb{R}^2$-plane as a function $|\mathbb{D}^*| = f(|\mathbb{S}^*|)$; see Figure 1. Following [11], we know that for $n \geqslant -1/2$ or $n < -1/2$ and $\gamma \geq d_n$, with:

$$d_n = 2\left[\frac{|2n+1|}{2(1-n)}\right]^{1-n}, \tag{9}$$

the function $|\mathbb{D}^*| = f(|\mathbb{S}^*|)$ defined in (8) is monotonically increasing and thus invertible as $|\mathbb{S}^*| = f^{-1}(|\mathbb{D}^*|)$. On the other hand, if $n < -1/2$ and $\gamma < d_n$, the relation (8) is no longer monotone and cannot be rewritten in the form $|\mathbb{S}^*| = f^{-1}(|\mathbb{D}^*|)$. In Figure 1, we plot the constitutive relation (8) with $\alpha^* = 1$, $\beta^* = 2$, and $\gamma = 0.1$ for different values of the index $n$. As one can see, the monotonicity is lost for $n = -5, \ -3, \ -2, \ -1$ for which $d_n > \gamma$.

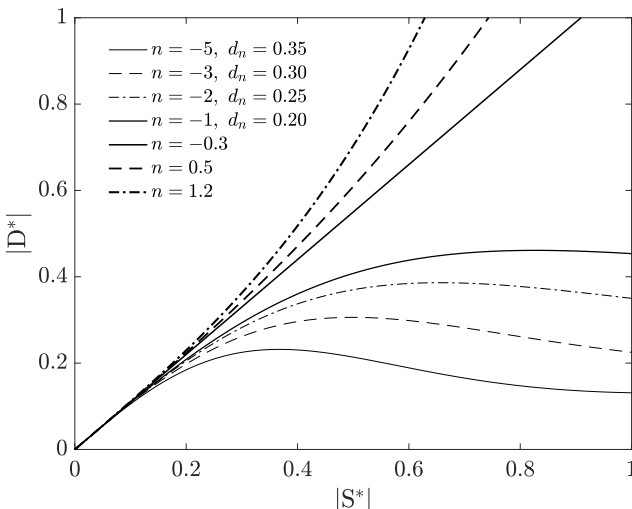

**Figure 1.** Constitutive relation (8) with $\alpha^* = 1$, $\beta^* = 2$, and $\gamma = 0.1$. The relation is monotone as long as $n \geqslant -1/2$ or $n < -1/2$ and $\gamma \geq d_n$.

In this paper, we studied the squeeze flow between parallel circular disks of a fluid whose constitutive equation is of the type of (7) under the assumption that the gap between the squeezing disks is much smaller than their radius (lubrication flow; see Figure 2). Squeeze flows are found in many engineering, biological, and rheological applications. In the context of biomechanical valves and diarthrodial joints are examples of squeeze flows relevant in biology and bioengineering. Other examples can be found in the context of food intake (chewing) where a material is squeezed between rigid surfaces (teeth) and where the gap between the rigid plates is smaller than the characteristic length of the squeezing surfaces.

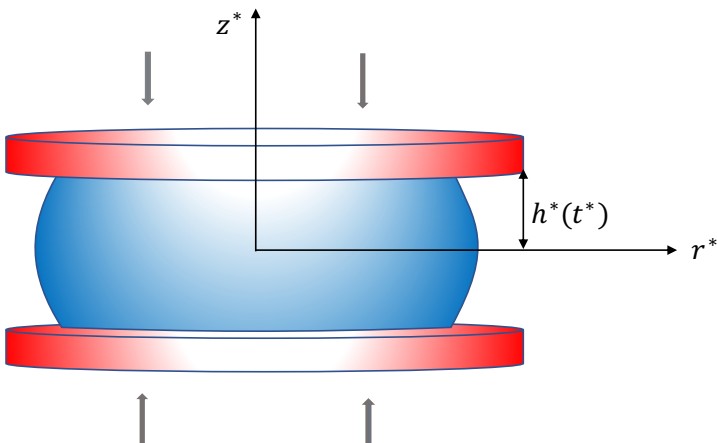

**Figure 2.** Sketch of the system.

For a general review on squeeze flow between disks of both Newtonian and non-Newtonian fluids, we refer the reader to [14–17]. We formulated the problem using cylindrical coordinates and assuming azimuthal symmetry so that all the variables do not depend on the angular coordinate. We looked for a solution that can be expressed

as power series of the small aspect ratio parameter $\varepsilon$ that represents the ratio between the half-distance that separates the disks and their radius. We focused on the leading- and first-order approximations for which we determined semi-analytical solutions for the velocity, stress, and pressure fields. We shall finally provide a graphical representation of the velocity, stress, and pressure fields, and we discuss the obtained results.

## 2. Squeeze Flow

We considered a portion of fluid squeezed between parallel disks of radius $R^*$ placed at a distance $2h^*(t^*)$; see Figure 2. We assumed that the plates squeezed the flow so that $dh^*/dt^* < 0$, and we also assumed azimuthal symmetry so that there was no dependence on the angular coordinate $\theta$. We defined $H^* = h^*(0)$ as the initial position of the upper plate and assumed that:

$$\varepsilon =: \frac{H^*}{R^*} \ll 1.$$

The velocity field is given by:

$$\boldsymbol{v}^* = v_r^*(r^*, z^*, t^*)\boldsymbol{e}_r + v_z^*(r^*, z^*, t^*)\boldsymbol{e}_z. \tag{10}$$

The constitutive equation is $\mathbb{T}^* = -p^*\mathbb{I} + \mathbb{S}^*$ where $\mathbb{S}^*$ is related to the symmetric part of the velocity gradient $\mathbb{D}^*$ via the implicit relation (7). The balance of mass and linear momentum in cylindrical coordinates yields:

$$\frac{\partial v_z^*}{\partial z^*} + \frac{1}{r^*}\frac{\partial}{\partial r^*}\left(r^* v_r^*\right) = 0, \tag{11}$$

$$\varrho^*\left(\frac{\partial v_r^*}{\partial t^*} + v_r^*\frac{\partial v_r^*}{\partial r^*} + v_z^*\frac{\partial v_r^*}{\partial z^*}\right) = -\frac{\partial p^*}{\partial r^*} + \frac{1}{r^*}\frac{\partial}{\partial r^*}\left(r^* S_{rr}^*\right) + \frac{\partial S_{rz}^*}{\partial z^*} - \frac{S_{\theta\theta}^*}{r^*}, \tag{12}$$

$$\varrho^*\left(\frac{\partial v_z^*}{\partial t^*} + v_r^*\frac{\partial v_z^*}{\partial r^*} + v_z^*\frac{\partial v_z^*}{\partial z^*}\right) = -\frac{\partial p^*}{\partial z^*} + \frac{1}{r^*}\frac{\partial}{\partial r^*}\left(r^* S_{rz}^*\right) + \frac{\partial S_{zz}^*}{\partial z^*}. \tag{13}$$

Exploiting the symmetry with respect to $z$, we write the boundary conditions as:

$$\begin{cases} v_z^* = \dfrac{dh^*}{dt^*}, \quad \lambda K^* S_{rz}^* + (1-\lambda)v_r^* = 0, \quad \text{on } z^* = h^*(t^*), \\[2mm] v_z^* = 0, \quad S_{rz}^* = 0, \quad\quad\quad\quad\quad\quad\quad\quad\quad \text{on } z^* = 0, \\[2mm] v_r^* = 0, \quad S_{rz}^* = 0, \quad\quad\quad\quad\quad\quad\quad\quad\quad \text{on } r^* = 0, \\[2mm] -p^* + S_{rr}^* = -p_{out}^* + \dfrac{\Gamma^*}{R^*}, \quad\quad\quad\quad\quad\quad\quad \text{on } r^* = R^*, \end{cases} \tag{14}$$

where $\Gamma^*$ represents the interfacial tension and $R^*$ is the sample radius, which is equivalent to the radius of curvature at the surface, assuming that the sample maintains a cylindrical shape. The quantity $p_{out}^*$ represents the external pressure at $r^* = R^*$. On the upper plate, the coefficient $\lambda \in [0,1]$ discriminates among different boundary conditions. Indeed, we observed that $\lambda = 0$ corresponds to the no-slip condition, $\lambda = 1$ represents free-slip, and $\lambda \in (0,1)$ stands for partial-slip with a friction coefficient that is related to the coefficient $K^*$.

We rescaled the system with the following non-dimensional variables:

$$r^* = R^* r, \quad z^* = H^* z, \quad t^* = \left(\frac{R^*}{U^*}\right)t, \quad p^* = p_{out}^* + \left(\frac{U^* R^*}{\alpha^* H^{*2}}\right)p, \tag{15}$$

$$v_r^* = U^* v_r, \quad v_z^* = \varepsilon U^* v_z, \quad \mathbb{S}^* = \left(\frac{U^*}{\alpha^* H^*}\right)\mathbb{S}, \quad \mathbb{D}^* = \left(\frac{U^*}{H^*}\right)\mathbb{D}, \quad \beta^* = \left(\frac{\alpha^* H^*}{U^*}\right)^2\beta, \tag{16}$$

so that (7) becomes:

$$\mathbb{D} = \left[ \left( 1 + \frac{\beta}{2} \operatorname{tr} \mathbb{S}^2 \right)^n + \gamma \right] \mathbb{S}, \tag{17}$$

while (11)–(13):

$$\frac{\partial v_z}{\partial z} + \frac{1}{r} \frac{\partial}{\partial r} (r v_r) = 0, \tag{18}$$

$$\varepsilon Re \left[ \frac{\partial v_r}{\partial t} + v_r \frac{\partial v_r}{\partial r} + v_z \frac{\partial v_r}{\partial z} \right] = -\frac{\partial p}{\partial r} + \frac{\varepsilon}{r} \frac{\partial}{\partial r} (r S_{rr}) + \frac{\partial S_{rz}}{\partial z} - \varepsilon \frac{S_{\theta\theta}}{r}, \tag{19}$$

$$\varepsilon^3 Re \left[ \frac{\partial v_z}{\partial t} + v_r \frac{\partial v_z}{\partial r} + v_z \frac{\partial v_z}{\partial z} \right] = -\frac{\partial p}{\partial z} + \frac{\varepsilon^2}{r} \frac{\partial}{\partial r} (r S_{rz}) + \varepsilon \frac{\partial S_{zz}}{\partial z}. \tag{20}$$

The non-dimensional boundary conditions acquire the form:

$$
\begin{cases}
v_z = \dot{h}(t), \quad \lambda K S_{rz} + (1 - \lambda) v_r = 0, \quad \text{on} \ \ z = h(t), \\[2mm]
v_z = 0, \quad S_{rz} = 0, \quad \text{on} \ \ z = 0, \\[2mm]
v_r = 0, \quad S_{rz} = 0, \quad \text{on} \ \ r = 0, \\[2mm]
-p + \varepsilon S_{rr} = \dfrac{\varepsilon^2}{Ca}, \quad \text{on} \ \ r = 1,
\end{cases}
\tag{21}
$$

where $\dot{h}(t) = dh/dt$ and where:

$$Ca = \left( \frac{U^*}{\Gamma^* \alpha^*} \right),$$

is the capillary number representing the relative effect of viscous drag forces versus surface tension forces acting across the interface between the liquid and the atmosphere at $r = 1$. In the following, we assumed that $Ca = O(1)$. We looked for a solution that could be expressed as a power series of $\varepsilon$, namely:

$$\boldsymbol{v}(r, z, t) = \boldsymbol{v}^{(0)}(r, z, t) + \varepsilon \boldsymbol{v}^{(1)}(r, z, t) + \varepsilon^2 \boldsymbol{v}^{(2)}(r, z, t) + \ldots \ldots \tag{22}$$

$$p(r, z, t) = p^{(0)}(r, z, t) + \varepsilon p^{(1)}(r, z, t) + \varepsilon^2 p^{(2)}(r, z, t) + \ldots \ldots \tag{23}$$

$$\mathbb{S}(r, z, t) = \mathbb{S}^{(0)}(r, z, t) + \varepsilon \mathbb{S}^{(1)}(r, z, t) + \varepsilon^2 \mathbb{S}^{(2)}(r, z, t) + \ldots \ldots \tag{24}$$

Using the expressions above, we could reformulate the mathematical problem for each order of approximation (leading, first, etc.). The system was therefore transformed into a series of simpler problems that could be solved iteratively. We focused on the leading- and first-order of approximation only, but it is easy to see that our procedure could be extended to any grade of approximation. We observed that from our scaling, the symmetric part of the velocity gradient is given by:

$$\mathbb{D} = \begin{bmatrix} \varepsilon \dfrac{\partial v_r}{\partial r} & 0 & \dfrac{1}{2} \left( \dfrac{\partial v_r}{\partial z} + \varepsilon^2 \dfrac{\partial v_z}{\partial r} \right) \\[4mm] 0 & \varepsilon \dfrac{v_r}{r} & 0 \\[4mm] \dfrac{1}{2} \left( \dfrac{\partial v_r}{\partial z} + \varepsilon^2 \dfrac{\partial v_z}{\partial r} \right) & 0 & \varepsilon \dfrac{\partial v_z}{\partial z} \end{bmatrix}, \tag{25}$$

implying that $S_{r\theta} = S_{\theta z} = 0$. It is straightforward to prove that:

$$\mathbb{D} = \left[ \left( 1 + \beta \left( S_{rz}^2 + \frac{S_{rr}^2 + S_{\theta\theta}^2 + S_{zz}^2}{2} \right) \right)^n + \gamma \right] \mathbb{S}. \tag{26}$$

From (25) and (26), we immediately see that $S_{rr}^{(0)} = S_{\theta\theta}^{(0)} = S_{zz}^{(0)} = 0$, meaning that the normal stresses are negligible at the leading order and that the only non-zero component of the deviatoric stress is $S_{rz}^{(0)}$. Exploiting this fact and expansion (24), we found that the term in the square brackets in (26) could be rewritten expanding in Taylor series around $\varepsilon$:

$$\left[\left(1 + \beta S_{rz}^{(0)^2}\right)^n + \gamma\right] + \left[2n\beta S_{rz}^{(0)}\left(1 + \beta S_{rz}^{(0)^2}\right)^{n-1}\right]S_{rz}^{(1)}\varepsilon + o(\varepsilon). \tag{27}$$

The components of the tensor $\mathbb{D}$ up to the first order thus become:

$$\varepsilon\frac{\partial v_r^{(0)}}{\partial r} = \left[\left(1 + \beta S_{rz}^{(0)^2}\right)^n + \gamma\right]\varepsilon S_{rr}^{(1)} \tag{28}$$

$$\varepsilon\frac{v_r^{(0)}}{r} = \left[\left(1 + \beta S_{rz}^{(0)^2}\right)^n + \gamma\right]\varepsilon S_{\theta\theta}^{(1)}, \tag{29}$$

$$\varepsilon\frac{\partial v_z^{(0)}}{\partial z} = \left[\left(1 + \beta S_{rz}^{(0)^2}\right)^n + \gamma\right]\varepsilon S_{zz}^{(1)}, \tag{30}$$

$$\frac{1}{2}\frac{\partial v_r^{(0)}}{\partial z} + \frac{\varepsilon}{2}\frac{\partial v_r^{(1)}}{\partial z} = \left[\left(1 + \beta S_{rz}^{(0)^2}\right)^n + \gamma\right]S_{rz}^{(0)} +$$
$$\varepsilon\left[\left(1 + \beta S_{rz}^{(0)^2}\right)^{n-1}\left(1 + \beta S_{rz}^{(0)^2}(2n+1)\right) + \gamma\right]S_{rz}^{(1)} \tag{31}$$

Let us now see how to write the boundary conditions on $z = h(t)$. To be as general as possible, we assumed that $h$ depends on $\varepsilon$ so that expanding $h = h(t; \varepsilon)$ around $\varepsilon = 0$, we write:

$$h = h^{(0)}(t) + \varepsilon h^{(1)}(t) + \varepsilon^2 h^{(2)}(t) + \dotsc \tag{32}$$

and:

$$\boldsymbol{v}\left(r, h, t\right) = \boldsymbol{v}^{(0)}\left(r, h^{(0)} + \varepsilon h^{(1)} + o(\varepsilon), t\right) + \varepsilon\boldsymbol{v}^{(1)}\left(r, h^{(0)} + \varepsilon h^{(1)} + o(\varepsilon), t\right) + o(\varepsilon).$$

Expanding the right-hand side of the above in $\varepsilon$, we found:

$$\boldsymbol{v}\left(r, h, t\right) = \boldsymbol{v}^{(0)}\left(r, h^{(0)}, t\right) + \varepsilon\left[\boldsymbol{v}^{(1)}\left(r, h^{(0)}, t\right) + \frac{\partial\boldsymbol{v}^{(0)}}{\partial z}\left(r, h^{(0)}, t\right)h^{(1)}\right] + o(\varepsilon).$$

Hence, the first boundary condition in (21)$_1$ becomes:

$$v_z^{(0)}\left(r, h^{(0)}, t\right) + \varepsilon\left[v_z^{(1)}\left(r, h^{(0)}, t\right) + \frac{\partial v_z^{(0)}}{\partial z}\left(r, h^{(0)}, t\right)h^{(1)}\right] + o(\varepsilon) = \dot{h}^{(0)} + \varepsilon\dot{h}^{(1)} + o(\varepsilon).$$

Using the same argument, we found that the second boundary condition (21)$_1$ can be rewritten as:

$$\lambda K S_{rz}^{(0)}\left(r, h^{(0)}, t\right) + (1 - \lambda)v_r^{(0)}\left(r, h^{(0)}, t\right) +$$
$$\varepsilon\left[\lambda K\left(S_{rz}^{(1)}\left(r, h^{(0)}, t\right) + \frac{\partial S_{rz}^{(0)}}{\partial z}\left(r, h^{(0)}, t\right)h^{(1)}\right) +$$
$$(1 - \lambda)\left(v_r^{(1)}\left(r, h^{(0)}, t\right) + \frac{\partial v_r^{(0)}}{\partial z}\left(r, h^{(0)}, t\right)h^{(1)}\right)\right] + o(\varepsilon) = 0.$$

**Remark 1.** *When $h(t)$ is independent of $\varepsilon$, we obtained $h(t) \equiv h^{(0)}(t)$, and the boundary conditions on $z = h^{(0)}$ reduced to:*

$$\begin{cases} v_z^{(0)}\left(r,h^{(0)},t\right) = \dot{h}^{(0)}(t), \quad v_z^{(j)}\left(r,h^{(0)},t\right) = 0, \quad j > 0 \\[2mm] \lambda K S_{rz}^{(j)}\left(r,h^{(0)},t\right) + (1-\lambda)v_r^{(j)}\left(r,h^{(0)},t\right) = 0 \quad j \geqslant 0 \end{cases}$$

**Remark 2.** *The dimensional force per unit surface (stress) exerted by the fluid on the upper plate is given by:*

$$-\mathbb{T}^* e_z\Big|_{z^*=h^*} = -\Big[\underbrace{\left(\mathbb{T}^* e_z \cdot e_z\right)e_z}_{normal} + \underbrace{\left(\mathbb{T}^* e_z \cdot e_r\right)e_r}_{tangential}\Big]\Big|_{z^*=h^*} =$$

$$= \Big[\left(p^* - S_{zz}^*\right)e_z - \left(S_{rz}^*\right)e_r.\Big]\Big|_{z^*=h^*}.$$

*The normal force per unit surface acting on the plate is thus:*

$$\mathfrak{f}^*(r^*,t^*) = p^*(r^*,h^*,t^*) - S_{zz}^*(r^*,h^*,t^*).$$

*Rescaling $\mathfrak{f}^*$ as the pressure, we found.*

$$\mathfrak{f} = \left(\frac{\alpha^* H^* \varepsilon}{U^*}\right)\mathfrak{f}^* = p_{out} + p - \varepsilon S_{zz}, \quad p_{out}^* = \left(\frac{U^*}{\alpha^* H^* \varepsilon}\right)p_{out},$$

*The resultant non-dimensional normal force is thus:*

$$\mathcal{F}(t) = 2\pi \int_0^1 \mathfrak{f}(r,t)r dr.$$

*We shall see that $p^{(0)}$ and $p^{(1)}$ depend only on r and t so that, making use of the asymptotic expansion and recalling that $S_{zz}^{(0)} = 0$, we found that:*

$$\mathcal{F}^{(0)}(t) = 2\pi \int_0^1 \Big[p_{out} + p^{(0)}(r,t)\Big]r dr,$$

$$\mathcal{F}^{(1)}(t) = 2\pi \int_0^1 \Big[p^{(1)}(r,t)\Big]r dr.$$

## 3. Zero-Order Approximation

At the leading order, the problem is reduced to:

$$\begin{cases} \dfrac{\partial v_z^{(0)}}{\partial z} = -\dfrac{1}{r}\dfrac{\partial}{\partial r}(rv_r^{(0)}) \\[3mm] -\dfrac{\partial p^{(0)}}{\partial r} + \dfrac{\partial S_{rz}^{(0)}}{\partial z} = 0 \\[3mm] -\dfrac{\partial p^{(0)}}{\partial z} = 0 \end{cases} \tag{33}$$

The boundary conditions are:

$$\begin{cases} v_z^{(0)} = \dot{h}^{(0)}, \quad \lambda K S_{rz}^{(0)} + (1-\lambda)v_r^{(0)} = 0, \quad \text{on } z = h^{(0)}(t), \\[2mm] v_z^{(0)} = 0, \quad S_{rz}^{(0)} = 0, \quad \text{on } z = 0, \\[2mm] v_r^{(0)} = 0, \quad S_{rz}^{(0)} = 0, \quad \text{on } r = 0, \\[2mm] p^{(0)} = 0, \quad \text{on } r = 1, \end{cases} \tag{34}$$

We immediately realized that $p^{(0)} = p^{(0)}(r,t)$ and $S_{rz}^{(0)} = G^{(0)}(r,t)z$, where $G^{(0)} = \partial p^{(0)}/\partial r$. The only non-zero component of $\mathbb{S}^{(0)}$ is $S_{rz}^{(0)}$ so that from (31):

$$\frac{1}{2}\frac{\partial v_r^{(0)}}{\partial z} = \left[\left(1 + \beta G^{(0)^2} z^2\right)^n + \gamma\right]G^{(0)}z. \tag{35}$$

The integration of (35) provides the radial component of the velocity field.

*3.1. The Case $\lambda = 1$*

We noticed that when $\lambda = 1$, the pressure gradient vanished, $G^{(0)} \equiv 0$. This was due to the fact that perfect slip implies no change in the internal pressure as the fluid is squeezed. Hence, $p^{(0)} = 0$ because of the boundary condition (34)₄. Moreover, $v_r^{(0)} = v_r^{(0)}(r,t)$, so that, from (33)₁:

$$v_z^{(0)}(r,z,t) = -\frac{1}{r}\frac{\partial}{\partial r}\left(rv_r^{(0)}\right)z, \tag{36}$$

where we used Condition (34)₂. The imposition of Condition (34)₁ yields:

$$\dot{h}^{(0)} = -\frac{1}{r}\frac{\partial}{\partial r}\left(rv_r^{(0)}\right)h^{(0)} \qquad \longrightarrow \qquad v_r^{(0)}(r,t) = -\frac{\dot{h}^{(0)}r}{2h^{(0)}} \tag{37}$$

where we exploited Condition (34)₃. Finally, replacing $v_r^{(0)}$ into (36), we found:

$$v_z^{(0)}(z,t) = \frac{z\dot{h}^{(0)}}{h^{(0)}}. \tag{38}$$

Notice that in this case, the solution did not depend either on $n$ or $\beta$ and, hence, on the particular constitutive equation we considered.

*3.2. The Case $\lambda \in [0,1)$*

Let us now consider the case $\lambda \in [0,1)$. Recalling that $S_{rz}^{(0)} = G^{(0)}(r,t)z$, the boundary condition for $v_r$ on the upper plate is:

$$v_r^{(0)}\Big|_{h^{(0)}} = \frac{\lambda K G^{(0)} h^{(0)}}{(\lambda - 1)}.$$

The integration of (35) provides:

$$v_r^{(0)}(r,z,t) = \frac{\lambda K G^{(0)} h^{(0)}}{(\lambda - 1)} - 2\int_z^{h^{(0)}}\left[\left(1 + \beta G^{(0)^2}\xi^2\right)^n + \gamma\right]G^{(0)}\xi d\xi, \tag{39}$$

or equivalently:

$$v_r^{(0)} = G^{(0)}\left[\frac{\lambda K h^{(0)}}{\lambda - 1} - \underbrace{\frac{\left(1 + \beta G^{(0)^2} h^{(0)^2}\right)^{n+1} - \left(1 + \beta G^{(0)^2} z^2\right)^{n+1}}{G^{(0)^2}\beta(n+1)}}_{=:\mathcal{M}^{(0)}} - \gamma\left(h^{(0)^2} - z^2\right)\right], \tag{40}$$

for $n \neq -1$ and:

$$v_r^{(0)} = G^{(0)}\left[\frac{\lambda K h^{(0)}}{\lambda - 1} - \underbrace{\frac{\ln\left(1 + \beta G^{(0)^2} h^{(0)^2}\right) - \ln\left(1 + \beta G^{(0)^2} z^2\right)}{G^{(0)^2}\beta}}_{=:\mathcal{M}^{(0)}} - \gamma\left(h^{(0)^2} - z^2\right)\right], \tag{41}$$

for $n = -1$. We observed that:

$$\frac{\partial v_r^{(0)}}{\partial t} = \frac{\partial G^{(0)}}{\partial t}\frac{v_r^{(0)}}{G^{(0)}} + G^{(0)}\left[\left(\frac{\lambda K}{1-\lambda} - 2\gamma h^{(0)}\right)\dot{h}^{(0)} - \frac{d\mathcal{M}^{(0)}}{dt}\right], \tag{42}$$

where:

$$\frac{d\mathcal{M}^{(0)}}{dt} = -2\left[\frac{dG^{(0)}}{dt}\left(1 + \beta G^{(0)2}z^2\right)^n\left(n\beta G^{(0)2}z^2 - 1\right) + \left(1 + \beta G^{(0)2}h^{(0)2}\right)^n \cdot\right.$$

$$\left.\cdot\left(\frac{dG^{(0)}}{dt} - G^{(0)3}\beta h^{(0)}\frac{dh^{(0)}}{dt} - G^{(0)2}\beta h^{(0)}n\frac{d(G^{(0)}h^{(0)})}{dt}\right)\right] \Big/ \left[G^{(0)3}\beta(n+1)\right], \tag{43}$$

when $n \neq -1$, and:

$$\frac{d\mathcal{M}^{(0)}}{dt} = \frac{dG^{(0)}}{dt}\left[\frac{2\ln(1 + \beta G^{(0)2}z^2) - 2\ln(1 + \beta G^{(0)2}h^{(0)2})}{G^{(0)3}\beta} + \right.$$

$$\left.\frac{2}{G^{(0)}}\left(\frac{h^{(0)2}}{(1 + \beta G^{(0)2}h^{(0)2})} - \frac{z^2}{(1 + \beta G^{(0)2}z^2)}\right)\right] + \frac{2h^{(0)}\dot{h}^{(0)}}{(1 + \beta G^{(0)2}h^{(0)2})}. \tag{44}$$

when $n = -1$. Equation (42) is useful for the evaluation of the solution at the first order. Integrating the continuity equation between $z$ and $h^{(0)}$, we found:

$$v_z^{(0)}(r,z,t) = \dot{h}^{(0)} + \frac{1}{r}\frac{\partial}{\partial r}\left[rG^{(0)}\int_z^{h^{(0)}}\left(\frac{\lambda K h^{(0)}}{\lambda - 1} - \mathcal{M}^{(0)} - \gamma(h^{(0)2} - \xi^2)\right)d\xi\right] \tag{45}$$

After some calculations, we obtained:

$$v_z^{(0)} = \dot{h}^{(0)} + \frac{1}{r}\frac{\partial}{\partial r}\left[rG^{(0)}(h^{(0)} - z)\left(\frac{\lambda K h^{(0)}}{\lambda - 1} - \frac{\gamma(2h^{(0)2} - h^{(0)}z - z^2)}{3}\right)\right.$$

$$\left. - rG^{(0)}\int_z^{h^{(0)}}\mathcal{M}^{(0)}d\xi\right]. \tag{46}$$

Equations (40), (41), and (46) provide the expressions of the velocity components at the leading order. To determine the pressure gradient $G^{(0)}$, we imposed the symmetry conditions (34)$_2$ so that:

$$0 = \dot{h}^{(0)} + \frac{1}{r}\frac{\partial}{\partial r}\left[rG^{(0)}h^{(0)}\left(\frac{\lambda K h^{(0)}}{\lambda - 1} - \frac{2\gamma h^{(0)2}}{3}\right) - rG^{(0)}\int_0^{h^{(0)}}\mathcal{M}^{(0)}d\xi\right]. \tag{47}$$

Integrating the above between zero and $r$ and exploiting the symmetry condition $G^{(0)}(0,t) = 0$, we obtained:

$$0 = \dot{h}^{(0)}\frac{r}{2} + G^{(0)}h^{(0)2}\left(\frac{\lambda K}{\lambda - 1} - \frac{2\gamma h^{(0)}}{3}\right) - G^{(0)}\int_0^{h^{(0)}}\mathcal{M}^{(0)}d\xi = 0. \tag{48}$$

Equation (48) is a nonlinear equation in $G^{(0)}$ whose solution provides the pressure gradient $G^{(0)}(r,t)$.

**Remark 3.** *We observed that:*

$$\lim_{\beta \to 0^+}\mathcal{M}^{(0)} = h^{(0)2} - z^2, \tag{49}$$

*irrespective of n and $G^{(0)}$, so that:*

$$G^{(0)}(r,t) = \frac{\dfrac{dh^{(0)}}{dt}\dfrac{r}{2}}{h^{(0)^2}\left[\dfrac{\lambda K}{1-\lambda} + \dfrac{2h^{(0)}(\gamma+1)}{3}\right]} = \frac{\partial p^{(0)}}{\partial r} < 0. \tag{50}$$

*Integrating between r and 1 and imposing $p^{(0)} = 0$ on $r = 1$, we found:*

$$p^{(0)}(r,t) = \frac{\dfrac{dh^{(0)}}{dt}\left(\dfrac{r^2-1}{4}\right)}{h^{(0)^2}\left[\dfrac{\lambda K}{1-\lambda} + \dfrac{2h^{(0)}(\gamma+1)}{3}\right]} \geqslant 0, \qquad r \in [0,1]. \tag{51}$$

*Equation (51) shows that our model made sense only if $(dh^{(0)}/dt)/h^{(0)^2}$ remained bounded in the time interval considered. If after some time, this was no longer valid, then the asymptotic expansion lost its validity since the pressure field within the fluid became too large. It is easy also to check that in the Newtonian case ($\beta \to 0$ and $\gamma = 0$):*

$$v_r^{(0)} = \frac{\dot{h}^{(0)}r\left[\dfrac{\lambda K h^{(0)}}{\lambda-1} - (h^{(0)^2} - z^2)\right]}{2h^{(0)^2}\left[\dfrac{\lambda K}{1-\lambda} + \dfrac{2h^{(0)}}{3}\right]}$$

$$v_z^{(0)} = \dot{h}^{(0)}\left[1 - \frac{h^{(0)} - z}{h^{(0)^2}\left(\dfrac{\lambda K}{1-\lambda} + \dfrac{2h^{(0)}}{3}\right)}\left(\dfrac{h^{(0)}\lambda K}{1-\lambda} + \dfrac{2h^{(0)^2} - h^{(0)}z - z^2}{3}\right)\right]$$

*The general case (47) does not provide an explicit expression for $G^{(0)}$, and the pressure gradient must be determined numerically.*

## 4. First-Order Approximation

At the first order, the problem is:

$$\begin{cases} \dfrac{\partial v_z^{(1)}}{\partial z} = -\dfrac{1}{r}\dfrac{\partial}{\partial r}\left(rv_r^{(1)}\right), \\[3mm] Re\left[\dfrac{\partial v_r^{(0)}}{\partial t} + v_r^{(0)}\dfrac{\partial v_r^{(0)}}{\partial r} + v_z^{(0)}\dfrac{\partial v_r^{(0)}}{\partial z}\right] = -\dfrac{\partial p^{(1)}}{\partial r} + \dfrac{\partial S_{rz}^{(1)}}{\partial z}, \\[3mm] -\dfrac{\partial p^{(1)}}{\partial z} = 0, \end{cases} \tag{52}$$

implying that the pressure is again independent of $z$, i.e., $p^{(1)} = p^{(1)}(r,t)$. The boundary conditions are:

$$\begin{cases} v_z^{(1)} = \dot{h}^{(1)} - \dfrac{\partial v_z^{(0)}}{\partial z} h^{(1)} & \text{on } z = h^{(0)}(t), \\[4mm] \lambda K S_{rz}^{(1)} + (1-\lambda)v_r^{(1)} = -h^{(1)}\left[\lambda K \dfrac{\partial S_{rz}^{(0)}}{\partial z} + (1-\lambda)\dfrac{\partial v_r^{(0)}}{\partial z}\right], & \text{on } z = h^{(0)}(t), \\[4mm] v_z^{(1)} = 0, \qquad S_{rz}^{(1)} = 0, & \text{on } z = 0, \\[4mm] v_r^{(1)} = 0, \qquad S_{rz}^{(1)} = 0, & \text{on } r = 0, \\[4mm] p^{(1)} = 0, & \text{on } r = 1. \end{cases} \tag{53}$$

From the definition of the stress components (28)–(30), we found:

$$S_{rr}^{(1)}\left[\left(1+\beta S_{rz}^{(0)2}\right)^n + \gamma\right] = \frac{\partial v_r^{(0)}}{\partial r}, \tag{54}$$

$$S_{\theta\theta}^{(1)}\left[\left(1+\beta S_{rz}^{(0)2}\right)^n + \gamma\right] = \frac{v_r^{(0)}}{r}, \tag{55}$$

$$S_{zz}^{(1)}\left[\left(1+\beta S_{rz}^{(0)2}\right)^n + \gamma\right] = \frac{\partial v_z^{(0)}}{\partial z}, \tag{56}$$

$$S_{rz}^{(1)}\underbrace{2\left[\left(1+\beta S_{rz}^{(0)2}\right)^{n-1}\left(1+\beta S_{rz}^{(0)2}(2n+1)\right) + \gamma\right]}_{=g^{(0)}(r,z,t)} = \frac{\partial v_r^{(1)}}{\partial z}, \tag{57}$$

showing that the normal stresses do not vanish at the first order. Let us now introduce:

$$f^{(0)}(r,z,t) = Re\left[\frac{\partial v_r^{(0)}}{\partial t} + v_r^{(0)}\frac{\partial v_r^{(0)}}{\partial r} + v_z^{(0)}\frac{\partial v_r^{(0)}}{\partial z}\right], \tag{58}$$

which is a function that is known from the leading-order problem. Notice that $f^{(0)}$ requires the knowledge of the time derivative of $v_r^{(0)}$, whose explicit form is given in (42). Integrating the momentum equation with the boundary condition $S_{rz}^{(1)} = 0$ on $z = 0$, we found:

$$\frac{\partial v_r^{(1)}}{\partial z} = g^{(0)}(r,z,t)\left[G^{(1)}(r,t)z + \int_0^z f^{(0)}(r,\xi,t)d\xi\right] = g^{(0)}(r,z,t)S_{rz}^{(1)}, \tag{59}$$

where $G^{(1)} = \partial p^{(1)}/\partial r$. Again, we distinguished between the free-slip condition and partial-/no-slip condition on $z = h^{(0)}$.

### 4.1. The Case $\lambda = 1$

When $\lambda = 1$, it is easy to check that $S_{rz}^{(1)} = 0$ on $z = h^{(0)}$ and that:

$$f^{(0)}(r,z,t) = Re\frac{r}{2}\left[\frac{3}{2}\left(\frac{\dot{h}^{(0)}}{h^{(0)}}\right)^2 - \frac{\ddot{h}^{(0)}}{h^{(0)}}\right],$$

As a consequence, $S_{rz}^{(1)}$ is linear in $z$, and since $S_{rz}^{(1)} = 0$ on $z = h^{(0)}$ and $z = 0$, we see that $S_{rz}^{(1)} \equiv 0$. From (59), we obtained:

$$G^{(1)}(r,t) = -Re\frac{r}{2}\left[\frac{3}{2}\left(\frac{\dot{h}^{(0)}}{h^{(0)}}\right)^2 - \frac{\ddot{h}^{(0)}}{h^{(0)}}\right], \tag{60}$$

since $g^{(0)}(r,z,y) = 2(1+\gamma)$ (recall that $S_{rz}^{(0)} \equiv 0$ when $\lambda = 1$). The pressure gradient at the first order does not vanish (as it does at order zero). Equation (60) provides the pressure gradient in the fluid in the case of perfect slip at the first order. Setting $S_{rz}^{(1)} = 0$ in (59), we obtained $v_r^{(1)} = v_r^{(1)}(r,t)$, so that, integrating the mass balance at the first order between zero and $z$ and recalling that $v_z^{(1)} = 0$ on $z = 0$, we found:

$$v_z^{(1)}(r,z,t) = -\frac{z}{r}\frac{\partial}{\partial r}\left[r v_r^{(1)}(r,t)\right]. \tag{61}$$

Now, we imposed the boundary condition $(53)_1$ that, recalling (38), can be rewritten as:

$$v_z^{(1)}(r,h^{(0)},t) = h^{(0)}\frac{d}{dt}\left(\frac{h^{(1)}}{h^{(0)}}\right).$$

Evaluating (61) on $z = h^{(0)}$ and integrating in $r$, we obtained:

$$\frac{r^2}{2}\frac{d}{dt}\left(\frac{h^{(1)}}{h^{(0)}}\right) + r v_r^{(1)}(r,t) = C(t), \qquad \Longrightarrow \qquad C(t) \equiv 0. \tag{62}$$

Therefore:

$$v_r^{(1)}(r,t) = -\frac{r}{2}\frac{d}{dt}\left(\frac{h^{(1)}}{h^{(0)}}\right), \tag{63}$$

while:

$$v_z^{(1)}(z,t) = z\frac{d}{dt}\left(\frac{h^{(1)}}{h^{(0)}}\right). \tag{64}$$

The solution at the first order is therefore completely determined by the solution at the leading order. Notice that, as it occurs at the leading order, the radial velocity is linear in $r$ and does not depend on $z$, whereas the vertical velocity is linear in $z$ and does not depend on $r$. Recalling (37), (38), and (54)–(57), we found:

$$S_{zz}^{(1)} = -2S_{rr}^{(1)} = -2S_{\theta\theta}^{(1)} = \frac{\dot{h}^{(0)}}{(1+\gamma)h^{(0)}}, \qquad S_{rz}^{(1)} \equiv 0, \tag{65}$$

which shows that, when $\lambda = 1$, the normal stresses depend only on time at the first order and that the shear stress is identically null.

*4.2. The Case $\lambda \in [0,1)$*

We started observing that:

$$\frac{\partial S_{rz}^{(0)}}{\partial z} = G^{(0)}(r,t), \qquad \frac{\partial v_r^{(0)}}{\partial z} = 2\left[(1 + \beta G^{(0)2}z^2)^n + \gamma\right]G^{(0)}(r,t)z$$

$$\frac{\partial v_z^{(0)}}{\partial z} = -\frac{1}{r}\frac{\partial}{\partial r}\left[rG^{(0)}(r,t)\left(\frac{\lambda K h^{(0)}}{\lambda - 1} - \mathcal{M}^{(0)} - \gamma(h^{(0)2} - z^2)\right)\right],$$

so that the boundary conditions $(53)_{1,2}$ became:

$$v_z^{(1)}\Big|_{h^{(0)}} = \dot{h}^{(1)} + \left[\frac{h^{(0)}h^{(1)}\lambda K}{\lambda - 1}\right]\frac{1}{r}\frac{\partial}{\partial r}\left(rG^{(0)}\right),$$

$$\left[\lambda K S_{rz}^{(1)} + (1-\lambda)v_r^{(1)}\right]\Big|_{h^{(0)}} = -\lambda K G^{(0)}h^{(1)}$$

$$+2(\lambda - 1)h^{(1)}\left[(1 + \beta G^{(0)2}h^{(0)2})^n + \gamma\right]G^{(0)}h^{(0)} = h^{(1)}\mathfrak{K}^{(0)}(r,t).$$

Recalling that:

$$S_{rz}^{(1)}(r,z,t) = \left[ G^{(1)}(r,t)z + \int_0^z f^{(0)}(r,\xi,t)d\xi \right],\tag{66}$$

we found that the boundary condition on $z = h^{(0)}$ for the velocity $v_r^{(1)}$ is given by:

$$v_r^{(1)}\bigg|_{h^{(0)}} = \frac{\lambda K\left[ G^{(1)}(r,t)h^{(0)} + \int_0^{h^{(0)}} f^{(0)}(r,\xi,t)d\xi \right] - h^{(1)}\mathfrak{K}^{(0)}(r,t)}{\lambda - 1},$$

where $G^{(1)}(r,t)$ is unknown. Therefore, integrating (59) between $z$ and $h^{(0)}$, we obtained:

$$v_r^{(1)}(r,z,t) = \frac{\lambda K\left[ G^{(1)}(r,t)h^{(0)} + \int_0^{h^{(0)}} f^{(0)}(r,\xi,t)d\xi \right] - h^{(1)}\mathfrak{K}^{(0)}(r,t)}{\lambda - 1}$$

$$- \int_z^{h^{(0)}} g^{(0)}(r,\xi,t)\left[ G^{(1)}(r,t)\xi + \int_0^{\xi} f^{(0)}(r,\eta,t)d\eta \right]d\xi.\tag{67}$$

Now, we integrated the mass balance between zero and $z$, obtaining:

$$v_z^{(1)} = -\frac{1}{r}\frac{\partial}{\partial r}\left[ r\int_0^z v_r^{(1)}(r,\xi,t)d\xi \right].\tag{68}$$

Then, we evaluated $v_z^{(1)}$ on $z = h^{(0)}$ and integrated in $r$:

$$\frac{r^2\dot{h}^{(1)}}{2} + \left[ \frac{h^{(0)}h^{(1)}\lambda K}{\lambda - 1} \right]rG^{(0)} + r\int_0^h v_r^{(1)}(r,\xi,t)d\xi = C(t), \qquad \Longrightarrow \qquad C(t) \equiv 0,$$

so that:

$$\frac{r\dot{h}^{(1)}}{2} + \underbrace{\left[ \frac{h^{(0)}h^{(1)}\lambda K}{\lambda - 1} \right]G^{(0)}}_{=h^{(1)}\mathfrak{M}^{(0)}(r,t)} + \int_0^{h^{(0)}} v_r^{(1)}(r,\xi,t)d\xi = 0.$$

Exploiting (67), we found:

$$\frac{\lambda Kh^{(0)}\left[ G^{(1)}(r,t)h^{(0)} + \int_0^{h^{(0)}} f^{(0)}(r,\eta,t)d\eta \right] - h^{(1)}h^{(0)}\mathfrak{K}^{(0)}(r,t)}{\lambda - 1} + h^{(1)}\mathfrak{M}^{(0)}(r,t) +$$

$$\frac{r\dot{h}^{(1)}}{2} = \int_0^{h^{(0)}} d\theta \int_\theta^{h^{(0)}} g^{(0)}(r,\xi,t)\left[ G^{(1)}(r,t)\xi + \int_0^{\xi} f^{(0)}(r,\eta,t)d\eta \right]d\xi,$$

from which we derived the explicit expression of $G^{(1)}(r,t)$:

$$G^{(1)}(r,t) = \left[ \frac{\lambda Kh^{(0)}\int_0^{h^{(0)}} f^{(0)}(r,\eta,t)d\eta - h^{(1)}h^{(0)}\mathfrak{K}^{(0)}(r,t)}{\lambda - 1} + h^{(1)}\mathfrak{M}^{(0)}(r,t) + \frac{r\dot{h}^{(1)}}{2} \right.$$

$$\left. - \int_0^{h^{(0)}} d\theta \int_\theta^{h^{(0)}} \left[ g^{(0)}(r,\xi,t)\int_0^{\xi} f^{(0)}(r,\eta,t)d\eta \right]d\xi \right] \cdot$$

$$\cdot \left[ \frac{\lambda Kh^{(0)2}}{1 - \lambda} + \int_0^{h^{(0)}} d\theta \int_\theta^{h^{(0)}} g^{(0)}(r,\xi,t)\xi d\xi \right]^{-1}.\tag{69}$$

Expression (69) finally allowed us to determine the velocity components (67) and (68). The stress $S_{rz}^{(1)}$ was obtained via (66) with $G^{(1)}$ given by (69).

**Remark 4.** *From the definition of $g^{(0)}$ and from the boundary condition $S_{rz}^{(0)} = 0$ on $r = 0$ and on $z = 0$, we see that:*

$$g^{(0)}(0, z, t) = g^{(0)}(r, 0, t) = 2(1 + \gamma) > 0.$$

*Therefore, a necessary condition for the denominator of (69) to vanish is that $g^{(0)}(r, z, t) = 0$ for some $r \in (0, 1]$, $t \geqslant 0$ and $z \in (0, h^{(0)}(t)]$. We proved here that when the constitutive relation (8) is monotone, i.e., $n \geqslant -1/2$ or $n < -1/2$ and $\gamma \geq d_n$, then the denominator of (69) is always strictly positive. Indeed, let us suppose that $g^{(0)}(r, z, t) = 0$ for some $(r, z) \in (0, 1] \times (0, h^{(0)}(t)]$ and $t \geqslant 0$. Then:*

$$\gamma = -\left(1 + \beta S_{rz}^{(0)^2}\right)^{n-1} \left[2n\beta S_{rz}^{(0)^2} + (1 + \beta S_{rz}^{(0)^2})\right]. \tag{70}$$

*Recalling that $\gamma$ is a non-negative parameter, the above equation has a solution only if the term in the square brackets of (70) is non-positive, i.e., only if:*

$$n \leqslant -\frac{(1 + \beta S_{rz}^{(0)^2})}{2\beta S_{rz}^{(0)^2}} < -\frac{1}{2}.$$

*Hence, a first necessary condition for $g^{(0)}$ to vanish is that $n < -1/2$ (if $n \geqslant 1/2$ the function $g^{(0)}(r, z, t)$ is always strictly positive). Now, let us suppose that $n < -1/2$ and set $X = \beta S_{rz}^{(0)^2} \geqslant 0$. Equation (70) becomes:*

$$\gamma = -(1 + X)^{n-1}\left[1 + X(2n + 1)\right] = m(X), \quad X \geqslant 0. \tag{71}$$

*It is straightforward to see that $m(0) = -1$ and $m(\infty) = 0$ (recall that we assumed that $n < -1/2$). Differentiating the function $m$ with respect to $X$, we obtained:*

$$m'(X) = -n(1 + X)^{n-2}\left[(2n + 1)X + 3\right],$$

*so that:*

$$m'(X_M) = 0 \quad \Longleftrightarrow \quad X_M = -\frac{3}{(2n + 1)} > 0,$$

*(the solution $X_M = -1$ is clearly disregarded). Now, we notice that:*

$$m(X_M) = 2\left[\frac{|2n + 1|}{2(1 - n)}\right]^{1-n} = d_n.$$

*Therefore, Equation (71) holds true only if $n < -1/2$ and $\gamma < d_n$, i.e., when the constitutive relation is non-monotone. In conclusion, we can state that, whenever the constitutive relation is monotonically increasing, the denominator in (69) is always strictly positive and the pressure gradient at the first order does not blow up.*

## 5. Results and Discussion

In this section, we show the contours/plots of the main variables involved in the problem for a given squeeze function $h(t)$. We did not have in mind any practical application; our intent was to give the reader an idea of the behavior of the system for different choices of the model parameters. We also observed that it was possible to extend our analysis to higher orders (second, third, etc.) to get a finer approximation of the solution. The procedure remained the same with the calculations becoming more involved.

To investigate the dependence on the physical parameters, we considered the following squeeze function:

$$h(t; \varepsilon) = \frac{\sqrt{1 + \varepsilon}}{\sqrt{1 + \varepsilon + t^2}} \approx \underbrace{\frac{1}{\sqrt{1 + t^2}}}_{h^{(0)}(t)} + \varepsilon \underbrace{\left[\frac{t^2}{2(1 + t^2)^{3/2}}\right]}_{h^{(1)}(t)} + o(\varepsilon), \tag{72}$$

so that:

$$\dot{h}(t;\varepsilon) = -\frac{t\sqrt{1+\varepsilon}}{(1+\varepsilon+t^2)^{3/2}} \approx \underbrace{-\frac{t}{(1+t^2)^{3/2}}}_{\dot{h}^{(0)}(t)} + \varepsilon \underbrace{\left[-\frac{t^3-2t}{2(1+t^2)^{5/2}}\right]}_{\dot{h}^{(1)}(t)} + o(\varepsilon). \tag{73}$$

The plot of the function $h(t;\varepsilon)$ together with its approximation is shown in Figure 3. Notice that $h(t;\varepsilon) > 0$ for all time $t \geqslant 0$, so that the plates cannot come in touch in a finite time. The values of the parameters appearing in the constitutive equation and in the boundary conditions are the ones of Table 1.

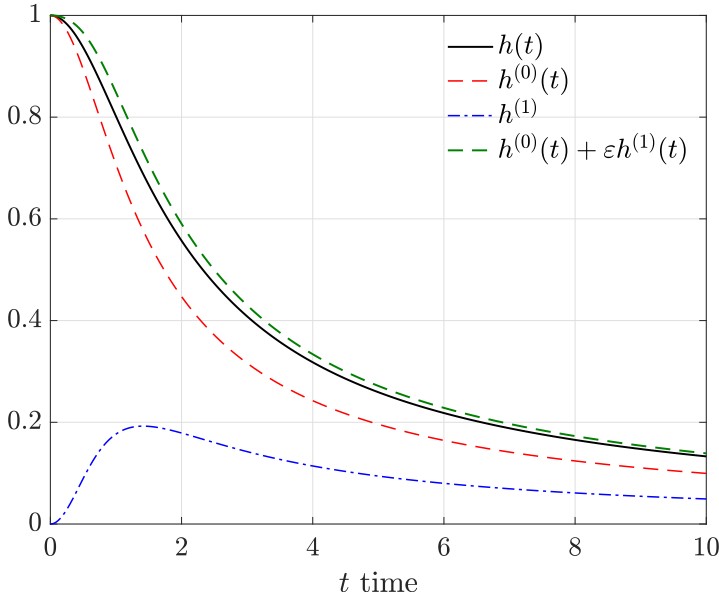

**Figure 3.** Plot of the function $h(t;\varepsilon)$ and its expansion $h^{(0)}(t) + \varepsilon h^{(1)}(t)$ with $\varepsilon = 0.3$.

**Table 1.** Parameters used for determining the contour plots of Figures 4 and 5.

| $\beta = 1$ | $n = 2$ | $\gamma = 1$ | $Re = 4$ | $K = 1$ | $\lambda = 0.2$ |
|---|---|---|---|---|---|

In Figure 4, we plot the contours of the velocity components, the contours of the tangential stress, and the flow pattern at the leading order. The parameters used in the simulations were the ones of Table 1. Time was set to $t = 0.5$ so that $h^{(0)} = 0.8944$. As physically expected, the modulus of the shear stress was maximum in the vicinity of the upper right corner of the $(r, z)$ domain, while it vanished on the axes $r = 0$ and $z = 0$. The radial velocity $v_r^{(0)}$ decreased with $z$ and increased with $r$, whereas the vertical velocity $v_z^{(0)}$ was almost uniform on $z = const$. In Figure 5, we show the velocities $v_r^{(1)}$, $v_z^{(1)}$ and the stress $S_{rz}^{(1)}$ still using the parameters of Table 1. We noticed that the radial velocity $v_r^{(1)}$ was negative (directed towards the center $r = 0$) except in the proximity of the axis $r = 0$ and near the plate. As a consequence, the radial velocity at the zero order was slightly larger than that obtained at the first order (the correction at the first order was negative almost everywhere). Recall that the first-order radial velocity is given by $v_r^{(0)} + \varepsilon v_r^{(1)}$.

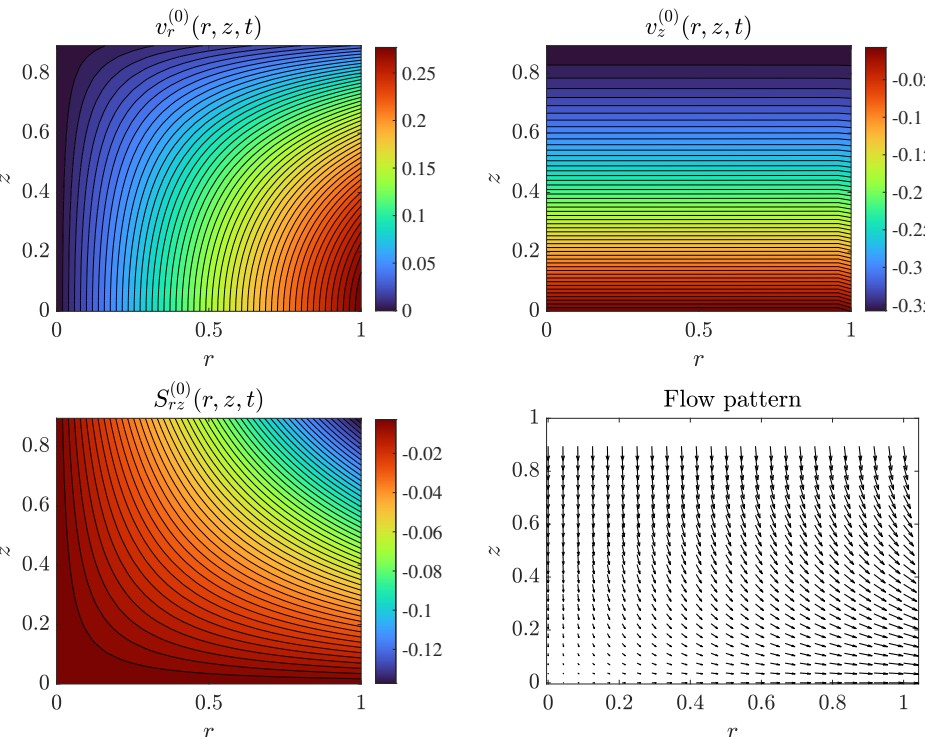

**Figure 4.** Contours of the velocity components $v_r^{(0)}$, $v_z^{(0)}$, stress $S_{rz}^{(0)}$, and flow patterns at the leading order. The contours were evaluated at time $t = 0.5$ with $h^{(0)}(0.5) = 0.8944$ and using the coefficients given in Table 1.

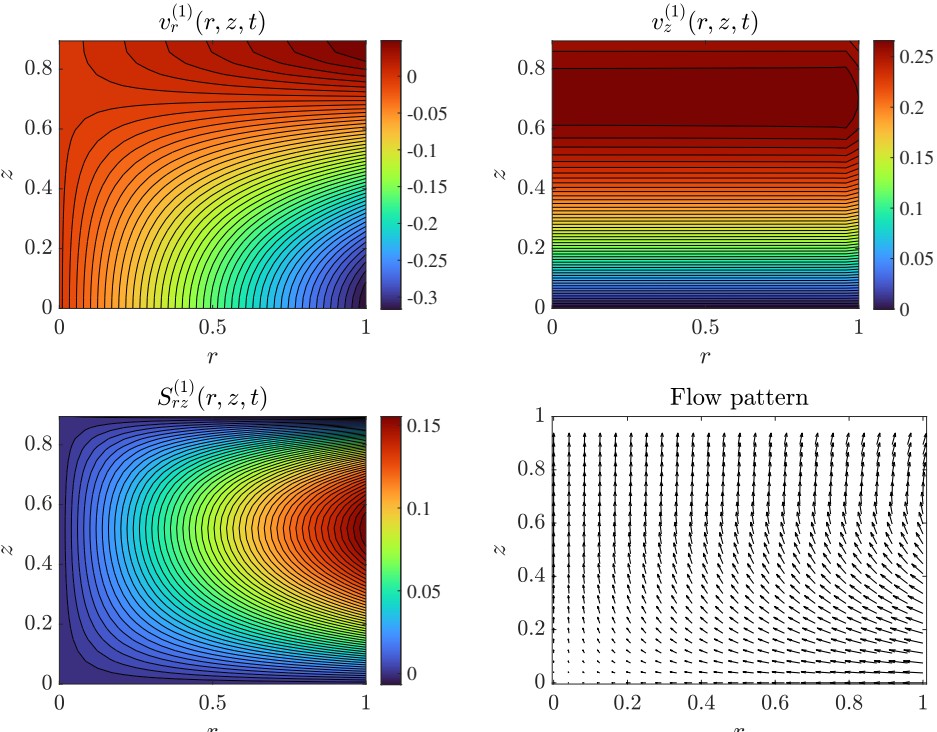

**Figure 5.** Contours of the velocity components $v_r^{(1)}$, $v_z^{(1)}$, stress $S_{rz}^{(1)}$, and flow patterns at the leading order. The contours were evaluated at time $t = 0.5$ with $h^{(0)}(0.5) = 0.8944$ and using the coefficients given in Table 1.

The stress $S_{rz}^{(1)}$ was positive almost everywhere and reached its maximum at the external surface $r = 1$, showing that the modulus of the shear stress at the first order was slightly smaller than the one at the zero order (recall that the shear stress at the leading order was negative everywhere). The normal stresses at the zero order were null, so we plot only $S_{rr}^{(1)}$, $S_{\theta\theta}^{(1)}$, and $S_{zz}^{(1)}$; see Figure 6. It is interesting to notice that the components $S_{rr}^{(1)}$ and $S_{\theta\theta}^{(1)}$ were identical. This was observed also for other choices of the parameters. Notice also that the moduli of the normal stresses were maximum on $z = 0$.

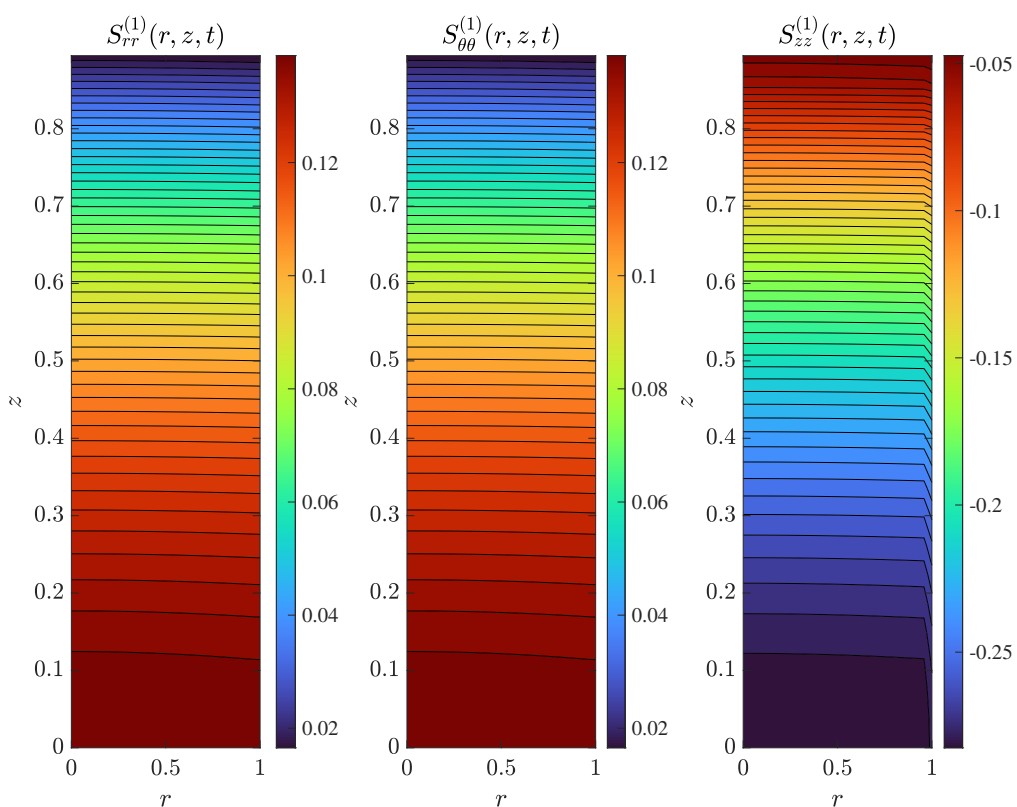

**Figure 6.** Contours of the stress components $S_{rr}^{(1)}$, $S_{\theta\theta}^{(1)}$, and $S_{zz}^{(1)}$. The contours were evaluated at time $t = 0.5$ with $h^{(0)}(0.5) = 0.8944$ and using the coefficients given in Table 1.

In Figure 7, we plot the shear stress, velocity components, and normal stresses at $r = 1$ and $t = 1$ (so that $h^{(0)}(1) = 0.7071$) for the indices $j = {}^{(0)}, {}^{(1)}$ and for $n = -5, -2, 0, 5$. The parameters here were $\lambda = 0$ (no-slip), $\gamma = 1$, $\beta = 3$, and $Re = 3$. These plots allowed us to study the dependence of the main variables on the index $n$. Looking at the shear stress plot, we noticed that $S_{rz}^{(0)}$ and $S_{rz}^{(1)}$ exhibited opposite monotonicity (with respect to $z$) irrespective of $n$. In particular, we noticed that at the leading order, the increase of $n$ resulted in a decrease in the modulus of the shear stress, so that the shear between adjacent layers was reduced. This was probably due to the fact that the fluid shear thinned as $n$ was increased. The velocity components $v_r^{(j)}$, $v_z^{(j)}$ for $j = {}^{(0)}, {}^{(1)}$ exhibited a less pronounced variation with respect to $n$. The ${}^{(0)}$ velocities were decreasing functions of $z$ with $v_z^{(0)}$ being negative for each $z$ (as expected). The normal stresses were monotonic functions of $z$ with $S_{rr}^{(1)}$ decreasing with $z$ and $S_{zz}^{(1)}$ increasing with $z$. We then illustrated the dependence of the pressure on $n$ at time $t = 1$. In Figure 8, we show the pressures $p^{(0)}$ and $p^{(1)}$ and the pressure at the first order $p^{(0)} + \varepsilon p^{(1)}$ at time $t = 1$. The parameters are the same as Figure 7. We noticed that whereas $p^{(0)}$ decreased with $n$ the pressure, $p^{(1)}$ actually increased with $n$ except for $n = 5$ (where the variation with $r$ was very small). The pressure at the leading order (green) decreased with $n$, showing that the correction $p^{(1)}$ did not vary the monotonicity of the pressure at the leading order. This behavior was again in

accordance with the fact that shear-thinning effects were enhanced as *n* increased. Notice finally that in all the plots, the Newtonian-like behavior was the one in which $n = 0$.

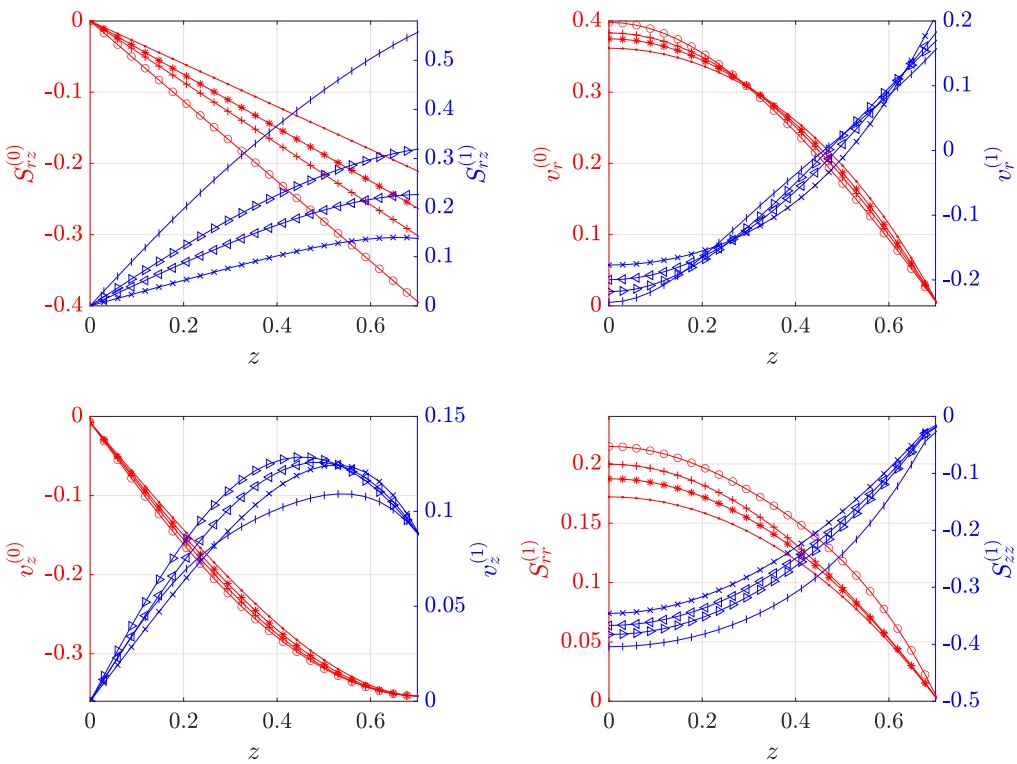

**Figure 7.** Shear stress $S_{rz}^{(j)}$, velocity components $v_r^{(j)}$, $v_z^{(j)}$ for $j = {}^{(0)}, {}^{(1)}$, and normal stresses $S_{rr}^{(1)}$, $S_{zz}^{(1)}$. The parameters are $\lambda = 0$ (no-slip), $\gamma = 1$, $\beta = 3$, and $Re = 3$: "o and |" ($n = -5$); "+ and ▷" ($n = -2$); "∗ and ◁" ($n = 0$), " and x" ($n = 5$).

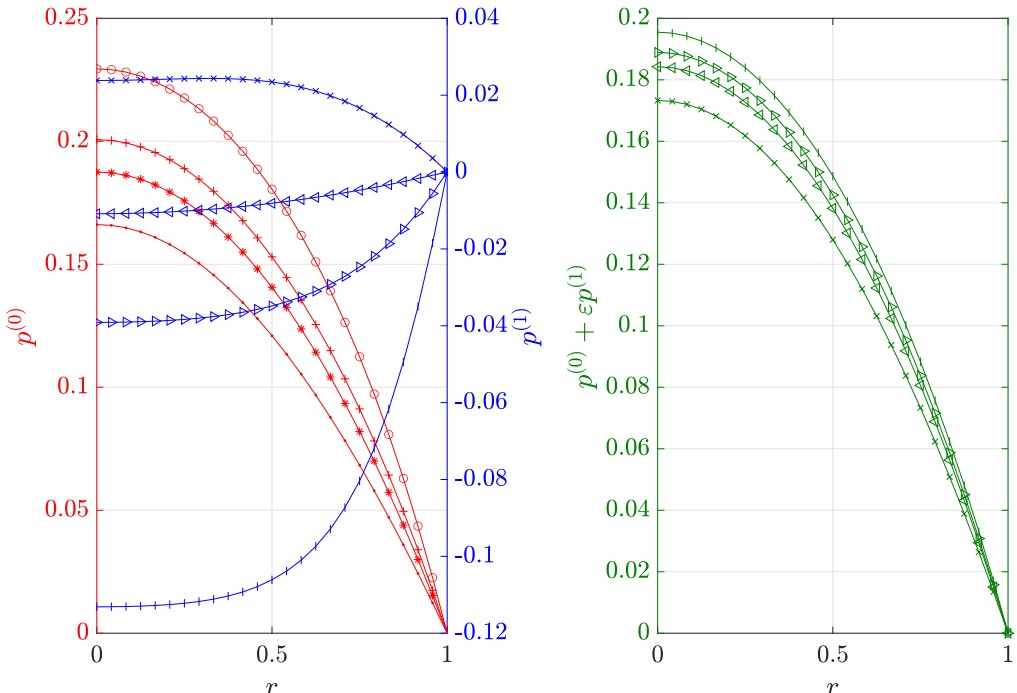

**Figure 8.** Pressure profiles for $\lambda = 0$, $\beta = 3$, $Re = 3$: o and | ($n = -5$); + and ▷ ($n = -2$); ∗ and ◁ ($n = 0$), and x ($n = 5$). On the right, the pressure at the first order, $\varepsilon = 0.3$.

Finally, in Figure 9, we show the total force exerted by the fluid on the plate; see Remark 2. We noticed that $\mathcal{F}^{(0)}$ increased with time, while $\mathcal{F}^{(1)}$ exhibited the opposite behavior. The total force at the first order $\mathcal{F}$ slightly increased at the beginning and then decreased with time. The correction at the first order in this case seemed to be quite relevant.

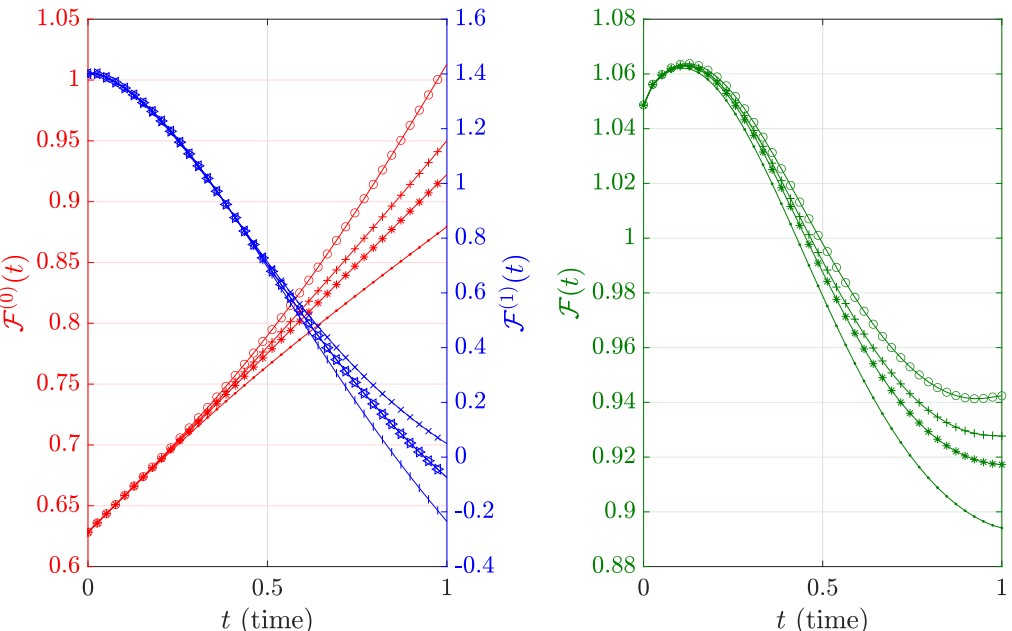

**Figure 9.** Total forces $\mathcal{F}^{(0)}(t)$, $\mathcal{F}^{(1)}(t)$, and $\mathcal{F}(t)$ exerted by the fluid on the plate as a function of time for different $n$ with $\lambda = 0$, $\beta = 3$, $Re = 3$, $p_{out} = 0.2$: 'o and |' $(n = -5)$; $+$ and $\triangleright$ $(n = -2)$; $*$ and $\triangleleft$ $(n = 0)$, and x $(n = 5)$. On the right, the total force at the first order, $\varepsilon = 0.3$.

The numerical results indicated that, within the context of squeeze flow, the deviation from the Newtonian behavior seemed to have a stronger effect on the stress distribution and on the pressure than on the velocity. This was probably due to the shear-thinning or shear-thickening nature of the stress power law model considered. In particular, we noticed that the pressure at the leading order could change by an order of magnitude with the flow index $n$. This is an important result if we think in terms of applications (think for instance at the synovial fluid between hyaline cartilage that can be modeled using lubrication approximation), since it demonstrated that the constitutive response of the fluid plays an important role in term of forces and stress acting within the fluid and on the squeezing surfaces.

## 6. Conclusions

We presented a mathematical model for the axisymmetric squeeze flow of a fluid whose constitutive equation is of an implicit type. We assumed that the geometrical setting was such that lubrication approximation could be used. We looked for a solution that was an asymptotic expansion in terms of the small aspect ratio parameter. We focused only on the zero- and first-order approximation, but the procedure adopted can be used to derive also higher order terms of the expansion (calculations become more involved as the order of approximation is increased). For the zero- and first-order problem, we determined the semi-explicit expression for the main variables, namely velocity, stress, pressure, and total force exerted by the fluid on the squeezing plates. We plotted the contours of the solution for different choices of the physical parameters, investigating in particular the behavior at the reference time at the interface between the fluid and the exterior. The graphical representations obtained can be used to compare the power stress law model with the Newtonian model that was obtained setting the flow index $n = 0$.

**Author Contributions:** Writing—original draft preparation, L.F. and A.B.; writing—review and editing, L.F. and A.B. All authors have read and agreed to the published version of the manuscript.

**Funding:** This research received no external funding.

**Acknowledgments:** The present work was performed under the auspices of the Italian National Group for Mathematical Physics (GNFM-Indam).

**Conflicts of Interest:** The authors declare no conflict of interest.

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
