# Peer review of "Squeeze Flow of Stress Power Law Fluids"

_fluids, doi:10.3390/fluids6060194_

Round 1
Reviewer 1 Report
Review of the manuscript Squeeze flow of stress power law fluids, by Fusi and Balloti. In this paper, the squeeze flow between disks of a non-Newtonian fluid is studied. The fluid rheology obeys the stress power law. The authors use a perturbation scheme to obtain an approximate solution of the hydrodynamic field. In the opinion of this reviewer, the manuscript is well written, and minor corrections are required. The paper could be accepted for publication in Fluids after the authors address the following points: 1. In the title: it should be “Squeeze flow of stress power-law fluids” 2. In the Introduction section, the authors must consider including more antecedents concerning the topic studied in the paper. The motivation for studying this problem must be emphasized. 3. The references must be ordered in the manuscript. Also, recent bibliography published in the last years about the topic regarded in the paper must be added. 4. Page 3, lines 45, 46: Justify under what conditions, from a physical and practical point of view, the aspect ratio is small. 5. Page 15, lines 92, 93: Although it is mentioned in the Results and discussion section that there is no practical application concerning the squeeze flow, It should be important to give insight about Eq. (72). There are many problems where the squeeze flow is applied. Please, consider focusing your analysis on some typical applications. 6. The Results and discussion section must be improved. The authors describe trends. However, the results must be explained based on fundamental aspects considering the squeeze flow problem, together with the rheology of the fluid.
Reviewer 2 Report
Comments on the manuscript “Squeeze flow of stress power law fluids” authored by Lorenzo Fuzzi and Andrea Ballotti
The Authors study a new class of constitutive fluids for which kinematical quantities are expressed as function of the stress, that is in the opposite way compared to the classical approach. They work on squeeze flows, of great importance for practical applications (including processing technologies for polymer materials, tribology, to name just two). The paper is very well written, clear, the results are rigorously presented. II recommend the manuscript be accepted in its present form.
Author Response
The replies to the referees are in the attached file